

# Orbicularis oculi muscle activity during computer reading under different degrees of artificially-induced aniseikonia

Beatriz Redondo[1,2], Jesus Vera[1,2], Rubén Molina[1], Alejandro Molina-Molina[3,4] and Raimundo Jiménez[1]

[1] Optics, Universidad de Granada, Granada, Granada, Spain
[2] New England College of Optometry, Boston, MA, United States of America
[3] Facultad de Ciencias de la Salud, Universidad San Jorge, Zaragoza, Spain
[4] Sport and Health University Research Institute (iMUDS), Department of Physical Education and Sports, Faculty of Sport Sciences, University of Granada, Granada, Spain

## ABSTRACT

**Background.** Aniseikonia is a binocular vision disorder that has been associated with asthenopic symptoms. However, asthenopia has been evaluated with subjective tests that make difficult to determine the level of aniseikonia. This study aims to objectively evaluate the impact of induced aniseikonia at different levels on visual fatigue by measuring the orbicularis oculi muscle activity in the dominant and non-dominant eyes while performing a reading task.

**Methods.** Twenty-four collegiate students (24.00 ± 3.86 years) participated in this study. Participants read a passage for 7 minutes under four degrees of aniseikonia (0%, 3%, 5% and 10%) at 50 cm. Orbicularis oculi muscle activity of the dominant and non-dominant eye was recorded by surface electromyography. In addition, visual discomfort was assessed after each task by completing a questionnaire.

**Results.** Orbicularis oculi muscle activity increased under induced aniseikonia (*i.e.*, greater values for the 10% condition in comparison to 0%, and 3% conditions ($p = 0.034$ and $p = 0.023$, respectively)). No statistically significant differences were observed in orbicularis oculi muscle activity for the time on task and between the dominant and non-dominant eyes. Additionally, higher levels of subjective visual discomfort were observed for lower degrees of induced aniseikonia.

**Conclusion.** Induced aniseikonia increases visual fatigue at high aniseikonia degrees as measured by the orbicularis oculi muscle activity, and at low degrees as measured with subjective questionnaires. These findings may be of relevance to better understand the visual symptomatology of aniseikonia.

Corresponding author
Jesus Vera, veraj@ugr.es

## INTRODUCTION

Aniseikonia is a binocular vision disorder in which the images of the two eyes are perceived as being different in size or shape. The prevalence of this condition is 1–3.5% (*Hughes, 1937*; *Burian, 1946*). The most common cause of aniseikonia is refractive magnification differences between the eyes due to anisometropia, which is either innate or acquired

(*i.e.,* cataract or corneal refractive surgery). Also, aniseikonia can be induced by the retina due to space alterations among the photoreceptors after stretching or compression of the retina (*Okamoto et al., 2017*), or it can be also associated with alterations in the visual cortex (*Bradley & Rabin, 1983*).

Aniseikonia is known to play an important role on quality of life, and humans are able to approximately tolerate 2% of aniseikonia before it becomes symptomatic (*Furr, 2019*). However, several clinical studies have described a high level of inter-subjects tolerance to aniseikonia, and the symptoms associated with this condition are highly variable across individuals (*Jiménez et al., 2002*; *Bharadwaj & Rowan Candy, 2011*; *Rutstein et al., 2015*; *Krarup et al., 2020*). In this regard, asthenopia and headaches are reported frequently, while diplopia, distorted space perception, impaired contrast sensitivity, photophobia, fatigue, nausea, vertigo, and dizziness are less frequent (*Rutstein, 1998*; *Furr, 2019*). Notably, aniseikonia affects visual performance, specifically, there is scientific evidence that 5% or higher degrees of induced aniseikonia deteriorate stereopsis (*Lovasik & Szymkiw, 1985*; *Jiménez et al., 2002*), binocular summation, contrast sensitivity (*Jiménez, Ponce & González-Anera, 2004*) and accommodation (*Jiménez et al., 2019*).

To date, the asthenopic symptoms associated with aniseikonia are based on the outcomes reported by participants after answering different vision-related questionnaires (*Hoffman, Girshick & Banks, 2008*; *Jiménez et al., 2019*; *Krarup et al., 2020*; *Tannous et al., 2021*). However, these subjective tests depends on subject's self-perception. The incorporation of surface electromyography recordings to assess the muscle activity from the orbital portion of the orbicularis oculi muscle (OO) has been proposed as an objective measure of asthenopia and visual fatigue in different contexts. For example, a number of asthenopia-inducing conditions such as glare (*Berman et al., 1994*; *Mork, Bruenech & Thorud, 2016*), visually demanding computer work (*Thorud et al., 2012*; *Yoo, 2014*), refractive error, (*Nahar et al., 2011*) low contrast stimuli, and small font size (*Gowrisankaran, Sheedy & Hayes, 2007*) have shown to alter the OO muscle activity.

In view of the lack of available scientific literature related to objective measures of aniseikonia strain, the present study aimed to objectively evaluate the impact of induced aniseikonia at different levels (0, 3, 5, and 10%) by measuring the OO muscle activity in the dominant and non-dominant eyes while performing a 7 min reading task. In addition, perceived levels of visual discomfort were also assessed by completing the questionnaire developed by *Hoffman, Girshick & Banks (2008)*. Based on previous studies (*Burian, 1946*; *Gowrisankaran, Sheedy & Hayes, 2007*; *Mork, Bruenech & Thorud, 2016*), we hypothesized that the OO muscular activity and visual discomfort will increase at higher levels of induced aniseikonia.

## METHODS

### Participants and ethical approval

Prior to data collection, we performed an a-priori power analysis with the G*Power 3.1.9.7 software assuming an effect size of 0.25, alpha of 0.05, and power 0.85, for a repeated measures (within factors) analysis of variance (ANOVA). The calculation projected

a required sample size of twenty participants. Twenty-four collegiate students (mean age ± standard deviation: 24.00 ± 3.86 years old; 16 female; (*Elliott, 2007*) −0.83 ± 0.57 D) were recruited to participate in this study. All participants accomplished the following inclusion criteria: (i) be free of any systemic ocular disease, (ii) have normal musculoskeletal health and be free of any neuromuscular disorder, (iii) visual acuity ≤ 0.0 log MAR in each eye with the best refractive correction, (iv) not presenting aniseikonia, as measured by the New Aniseikonia Test (Handaya Co., Tokyo, Japan) (*Awaya et al., 1982*), (v) being able to perceive the reading test with clarity and stable fusion in all cases of induced aniseikonia (0%, 3%, 5%, 10%), (vi) no history of strabismus, amblyopia, dry eye, refractive surgery, orthokeratology, binocular, accommodative or oculomotor disorders, (vii) belonging to the asymptomatic visual discomfort group based on the scores of the Conlon visual discomfort survey (cut off value <24), and (viii) not taking any medication. All participants were asked to abstain from alcohol and caffeine-based drinks 24 and 12 h before the experimental session, respectively, and to sleep at least 7 h the night prior to testing. OO muscle activity data from two participants were excluded due to recording errors, and thus, twenty-two participants were included for statistical analyses. The experimental protocol followed the guidelines of the Declaration of Helsinki and was approved by the University of Granada Institutional Review Board (IRB approval: 546/CEIH/2018). Written informed consent was obtained from all participants.

## Induced aniseikonia

Aniseikonia was artificially-induced by using afocal magnifiers (*Bharadwaj & Rowan Candy, 2011*; *Jiménez et al., 2019*) (Fig. 1). These lenses do not induce any change in defocus but rather magnify the image as the front and back surface optical powers cancel each other out (*Fannin & Grosvenor, 1996*). The afocal magnifier of 3%, 5% and 10% degrees of aniseikonia were individually placed in front of the sensory dominant eye (determined by judgment of stimulus contrast-polarity) (*Bossi et al., 2018*). These specific degrees were chosen since aniseikonia ranging from 2% to 5% is considered as the habitual limit of tolerance (*Howard, 1995*). Specifically, aniseikonia from 3–5% is associated with a reduction of stereopsis (*Vlaskamp, Filippini & Banks, 2009*), values greater than 5% of aniseikonia have shown to deteriorate binocular summation and contrast sensitivity (*Jiménez, Ponce & González-Anera, 2004*), and induced levels of aniseikonia higher than 8% cause a significant increase of the accommodative lag (*Jiménez et al., 2019*). Each lens was properly adjusted for the interpupillary distance and pupil heights using an adjustable half-eye trial frame. For the control condition (0%), no lens was placed in the trial frame. Participants wore soft contact lenses when necessary, since spectacle correction would induce aniseikonia during the reading task.

## Reading task

The reading task was generated using the PsychoPy2 software library (V.1.85.4) written in Phyton 3.6.3 (*Peirce, 2009*) and presented on a calibrated 15.6 in. LCD screen TV (B15A-PH, Oki Electric Industry Co., Ltd. Tokyo, Japan). Four different passages obtained from Spanish literature in the participant's native language (Spanish) were chosen. We

**Figure 1** A graphical illustration of the experimental design. Afocal magnifier of 3%, 5% and 10% degrees of aniseikonia.

checked the difficulty of the passages and ensured that all had a similar legibility according to the INFLESZ scale, with all the passages belonging to the category of "somewhat difficult" based on the classification of *Barrio-Cantalejo et al. (2008)*. In order to control that small differences in textual elements and difficulty among passages did not influence the effects of aniseikonia level on visual fatigue, the passage chosen for each condition was randomized across subjects. All passages had a duration longer than 7 minutes to ensure that participants did not finish the reading task before the established reading time (*i.e.,* 7 min). Verdana font type and a visual x-high angle of 0.22° were used in order to permit an optimal reading speed (*Sheedy, Smith & Hayes, 2005*; *Legge & Bigelow, 2011*). Subjects were seated at 50 cm from the screen with their head stabilized on the chin and forehead rest and aligned with the passage which was left-justified in a 40-characters per full-width format, and displayed in a window height of 15° at the screen center (*Kundart et al., 2010*). The initial vertical scrolling speed of text was adjusted to 0.25 cm/s, although participants were allowed to set the speed at which the text was scrolled on the screen by adjusting the arrows cursor of the computer keyboard throughout the duration of the reading task.

## Muscle activity measurements and signal processing

The OO activity signal was recorded from both eyes simultaneously while performing the reading task using the mDurance® System (mDurance Solutions SL, Granada, Spain), at a sampling rate of 1,024 Hz, following the procedure described by *Vera et al. (2022)*. The mDurance System is a digital validated tool that combines wearable surface electromyography, mobile computing and cloud analysis to streamline and automatize the assessment of muscle activity (*Molina-Molina et al., 2020*). The subject's skin was cleaned with a 75% alcohol-soaked cotton swab and was dried before the electrodes were placed. A total of five self-adhesive pre-gelled Ag/AgCl surface electrodes with a contact surface diameter of 10 mm were used. Two electrodes per eye were used, which were placed 1.5 cm below the lower eyelid margin midway between the medial and lateral canthi with an

inter-electrode distance of 20 mm. The reference electrode was placed on the forehead (Fig. 2). A baseline noise of less than 15 microvolts was verified prior to each recording. For the processing and filtering of the raw data, a fourth order Butterworth bandpass filter with a cut-off frequency of 20–450 Hz was used to remove low-frequency artifacts such as eye blinks or movements, motion potentials, activity of neighboring muscles, swallowing or respiration. Then, the signal was smoothed using a window size of 0.025 s root mean square (RMS) and an overlapping of 0.0125 s between windows for both systems separately. To normalize the RMS signal, the maximum voluntary isometric contraction was obtained for each eye. This value was obtained from the maximum OO muscle activity when participants were ask to close their eyes as hard as possible on three occasions. The main variable recorded for muscle activity was the mean RMS expressed in microV, whereby the higher the degrees, the greater the muscle activity. Normalization was performed by dividing the RMS signal during the reading task by the maximum voluntary isometric contraction, and this reference value was multiplied by 100 to be expressed as a percentage (*Burden, 2010*). The average value of the normalized OO muscle activity signal from each eye was used for statistical analyses.

## Subjective measures

Before and after performing the reading task, participants reported their subjective levels of visual fatigue and discomfort using the five-items questionnaire developed by *Hoffman, Girshick & Banks (2008)*. This questionnaire consists of five items ranging from 0 (no symptoms) to 4 (severe symptoms).

## Procedure

First, participants underwent an optometric examination to verify that the inclusion criteria were accomplished. All participants wore their habitual refractive correction. At this point, participants were given written and verbal instructions about the experimental conditions and the informed consent was signed. OO muscle activity was recorded while participants performed four 7 min reading tasks that only differed in the degree of aniseikonia (0%, 3%, 5% and 10%). The mDurance System only permits recording of muscle activity over a three minute block, and therefore the measurement points were the first and last three minutes (first block: 1–3 min; and second block: 4–7 min). Participants attended the laboratory on one occasion only, with the level of aniseikonia and the reading passages being chosen in a random manner with the Research Randomiser software (https://www.randomizer.org#randomize). The evaluation was conducted from 9.00 am to 1.00 pm. and it was never conducted just after eating. Five minutes breaks were given between two consecutive conditions. Immediately after each passage, subjective levels of visual discomfort were assessed. Also, three true-false questions to test their comprehension of the text were asked to ensure that participants were paying attention. Two correct answers were considered to ensure an adequate comprehension of the passage. The ambient illumination in the room, as measured in the corneal plane, was kept constant at ∼150 lux (Illuminance Meter T-10A, Konica Minolta, Tokyo, Japan).

                                                                                                     

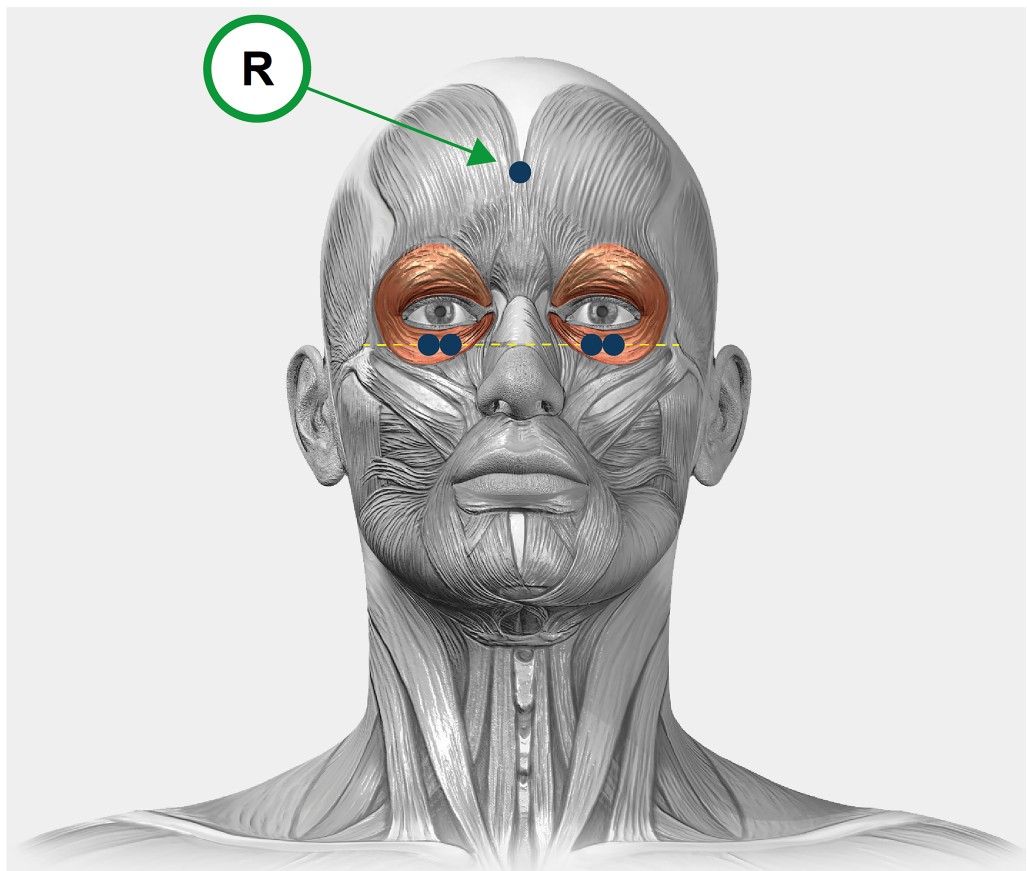

**Figure 2** **A graphical illustration of the experimental design.** Electrodes placement in the orbicularis oculi of the right and left eyes. The electrodes are represented in blue and the reference electrode is indicated with an "R".

## Experimental design and statistical analyses

We designed a repeated-measures study with the degree of induced aniseikonia (0, 3, 5, and 10%), the time on task (first block: 0–3 min, and second block: 4–7 min), and the eye dominance (dominant and non-dominant eye) as the within-participants factors, while the OO muscle activity and perceived levels of visual discomfort were the dependent variables.

Normal distribution of the data and the homogeneity of variances were confirmed by Shapiro–Wilk test and Levene's test, respectively ($p > 0.05$). For the main analysis, the OO muscle activity was submitted to a repeated measures ANOVA with the degree of induced aniseikonia (0, 3, 5, and 10%), the time on task (block 1 and block 2) and the eye dominance (dominant and non-dominant eye) as the within-participants factors. In addition, perceived levels of visual discomfort were analyzed using separate one-way repeated measures ANOVA with the degree of induced aniseikonia (0, 3, 5, and 10%) as the only within-participants factor. The magnitude of the differences was reported by the partial eta squared ($\eta_p^2$) for Fs and Cohen's d effect size (ES) for t-tests, respectively (*Cohen, 1988*). The criteria for interpreting the magnitude of the effect sizes were: small

**Table 1** Descriptive values (mean ± standard deviation) of the orbicularis oculi muscle activity measurement taken in this study.

| | Point of measure | Eye dominance | 0% aniseikonia | 3% aniseikonia | 5% aniseikonia | 10% aniseikonia |
|---|---|---|---|---|---|---|
| **Orbicularis oculi muscle activity (%)** | *First block* | *Dominant eye* | 10.1 ± 11.0 | 9.4 ± 8.3 | 12.2 ± 13.0 | 13.4 ± 10.0 |
| | | *Non-dominant eye* | 11.2 ± 8.9 | 11.5 ± 11.5 | 13.4 ± 13.5 | 14.3 ± 13.9 |
| | *Second block* | *Dominant eye* | 10.2 ± 11.9 | 9.2 ± 8.2 | 11.1 ± 11.6 | 13.9 ± 10.1 |
| | | *Non-dominant eye* | 10.6 ± 10.3 | 10.9 ± 11.8 | 14.7 ± 12.9 | 16.3 ± 13.5 |

(0.01), medium (0.06), and large (0.14) for eta squared (*Cohen, 1988*), and trivial (<0.2), small (0.2–0.6), moderate (0.6–1.2), large (1.2–2.0) and extremely large (>2.0) for Cohen's d (*Hopkins et al., 2009*). Statistical significance was set at 0.05, and multiple comparisons were corrected with the Holm-Bonferroni procedure. The JASP statistics package (version 0.16.2) was used for all statistical analyses.

## RESULTS

Table 1 shows the descriptive values of the OO muscle activity for the different experimental conditions. The analysis of the OO muscle activity exhibited a statistically significant effect for the main factor "aniseikonia level" ($F_{3,63} = 4.04$, $p = 0.011$, $\eta_p^2 = 0.16$), whereas the main factors of "time on task" and "eye dominance" did not reach statistical significance ($F_{1,21} = 0.11$, $p = 0.740$; and $F_{1,21} = 0.49$, $p = 0.491$; respectively). The interactive effects of "aniseikonia level × time on task" ($F_{3,63} = 0.61$, $p = 0.611$), "aniseikonia level × eye dominance" ($F_{3,63} = 0.14$, $p = 0.933$), "time on task × eye dominance" ($F_{1,21} = 0.47$, $p = 0.499$), and "aniseikonia level × time on task × eye dominance" ($F_{3,63} = 0.56$, $p = 0.646$) were not statistically significant.

*Post-hoc* analyses between the four aniseikonia levels exhibited greater values OO muscles activity for the 10% condition in comparison to the control (0%) and 3% conditions (corrected *p*-value = 0.034, Cohen's $d = 0.60$; and corrected *p*-value = 0.023, Cohen's $d = 0.64$; respectively). The rest of comparisons did not reach statistical significance (corrected *p*-values between 0.289 and 0.840) (Fig. 3).

Table 2 depicts the descriptive values for the five questions used. Separate unifactorial ANOVA analyses, considering the aniseikonia level as the only within-participants factor, exhibited a statistically significant effect for the aniseikonia level (*p*-values are reported in Table 2). Overall, higher levels of visual fatigue and discomfort were observed as a function of greater degrees of induced aniseikonia. *Post-hoc* analyses are included in Fig. 4.

## DISCUSSION

The present study showed that OO muscle activity increases as a function of the level of induced aniseikonia (*i.e.,* 10%). However, we did not observe an increase in the OO muscle activity as a function of time-on-task during the reading task. No statistically significant differences were observed in OO muscle activity between the dominant and non-dominant eyes. Also, perceived levels of visual discomfort were positively associated with the level of induced aniseikonia. This set of findings indicates that objective and subjective measures of visual fatigue are sensitive to aniseikonia.

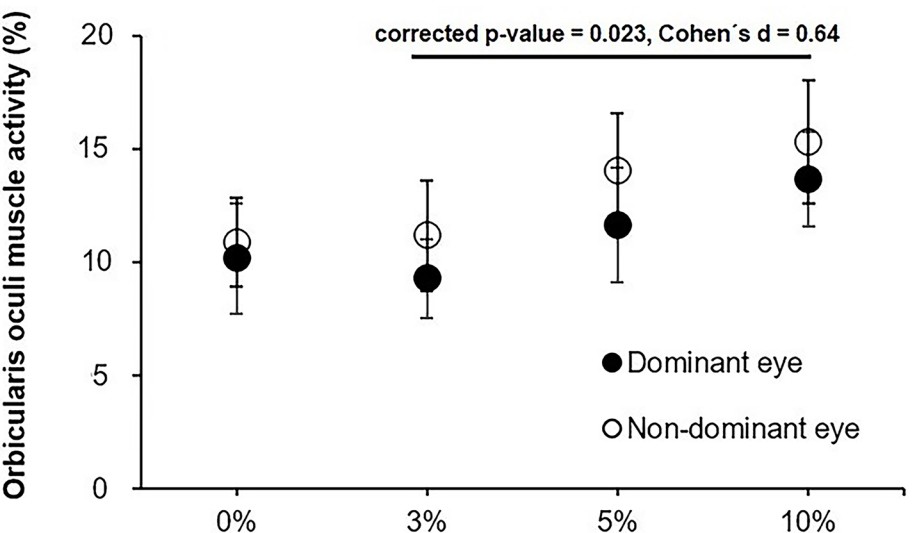

Figure 3 **Average orbicularis oculi muscle activity for the dominant and non-dominant eye for the 0, 3, 5 and 10% levels of induced aniseikonia ris oculi muscle activity for the dominant and non-dominant eye for the 0, 3, 5.** Error bars represent the standard error. Image source credit: mDURANCE.

Table 2 **Reported symptoms of visual fatigue and discomfort after completing the reading in the four aniseikonia conditions (0%, 3%, 5% and 10%) using the questionnaire developed by** *Hoffman, Girshick & Banks (2008)*.

| | 0% aniseikonia | 3% aniseikonia | 5% aniseikonia | 10% aniseikonia | P-value | F | $\eta_p^2$ |
|---|---|---|---|---|---|---|---|
| **Q1. How does your head feel? (0–4)** | $1.27 \pm 0.77$ | $1.50 \pm 0.91$ | $1.73 \pm 0.99$ | $1.86 \pm 0.94$ | 0.005 | 4.667 | 0.182 |
| **Q2. How does your eyes feel? (0–4)** | $0.96 \pm 0.84$ | $1.09 \pm 0.92$ | $1.36 \pm 0.85$ | $1.46 \pm 1.14$ | 0.012 | 3.921 | 0.157 |
| **Q3. How tired and sore are your neck and back? (0-4)** | $0.86 \pm 0.71$ | $1.00 \pm 0.82$ | $1.36 \pm 0.79$ | $1.32 \pm 1.09$ | 0.007 | 4.416 | 0.174 |
| **Q4. How clear is your vision? (0–4)** | $0.77 \pm 0.61$ | $1.18 \pm 0.85$ | $1.55 \pm 0.86$ | $1.64 \pm 1.05$ | <0.001 | 12.594 | 0.375 |
| **Q5. How tired are your eyes? (0–4)** | $0.50 \pm 0.51$ | $0.73 \pm 0.63$ | $0.96 \pm 0.79$ | $0.96 \pm 0.90$ | 0.010 | 4.070 | 0.162 |

Previous studies have reported that aniseikonia increases the perceived levels of asthenopia (*Rutstein, 1998*). In the same line, our results demonstrate that induced aniseikonia alters objective indices of visual fatigue, as evidenced by the greater OO muscle activity observed for the 10% level of induced aniseikonia in comparison to the control condition. As previously reported, this index has demonstrated to be sensitive

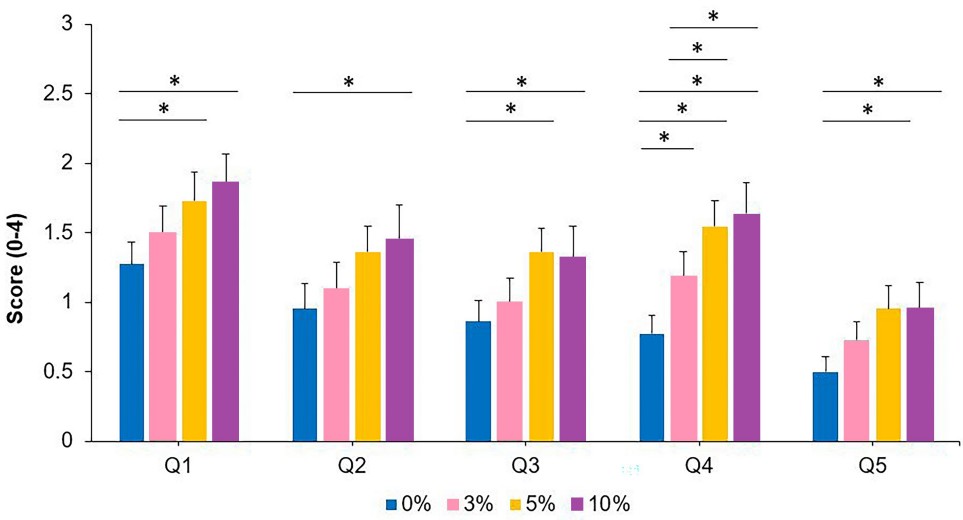

**Figure 4** **Symptoms of visual discomfort and fatigue for the 0, 3, 5 and 10% levels of induced aniseikonia for each of the five items of the** *Hoffman, Girshick & Banks (2008)* **questionnaire (Q1–Q5).** Scores range from 0 (no symptoms) to 4 (severe symptoms). *denotes statistically significant differences between the different levels of induced aniseikonia (corrected *p*-value < 0.05). Error bars represent the standard error.

to visual fatigue (*Gowrisankaran, Sheedy & Hayes, 2007*; *Nahar et al., 2007*; *Thorud et al., 2012*), and increasing OO muscle activity under artificially induced aniseikonia conditions may cause individuals to squint due to the visual distortion. It is plausible that participants were squeezing their eyes more in the presence of aniseikonia to gain clearer vision or reduce diplopia/ghosting. Future studies are needed to elucidate this possibility.

The increase in OO muscle activity was independent of the eye tested (*i.e.,* dominant *vs.* non-dominant), which could be explained by a binocular response to fuse both images. For its part, the changes in visual discomfort symptomatology observed in this study are in line with previous studies, which observed an effect of aniseikonia on visual symptomatology for 3 to 5% of aniseikonia (*Rutstein, 1998*; *Jiménez et al., 2019*). Interestingly, a considerably high degree of induced aniseikonia was needed to induce head, neck and back ache, and eyes strain and tiredness (5 and 10%), while blurriness was reported when using the 3% afocal magnifier.

It well known that OO muscle activity during a reading task increases under visually stressing conditions such as glare, low contrast text, small font size (*Gowrisankaran, Sheedy & Hayes, 2007*) and uncorrected refractive error (*Nahar et al., 2011*). Previous studies have examined whether OO muscle activity changed over time during a prolonged reading task, with these investigations obtaining mixed results. For example, *Thorud et al. (2012)* and *Vera et al. (2022)* in a 2 h and 30 min computer reading tasks, respectively, did not found any change in OO muscle activity, while *Mork, Bruenech & Thorud (2016)* during a 30 min visually stressing reading task (*i.e.,* glare) found an increase in OO muscle activity during the first minutes, but showed a decreasing trend after 15 to 20 min. Here, we did not find a significant change in OO muscle activity as a function of time-on-task, which

could be due to the relatively short duration of each reading condition. Future studies are needed to determine the impact of aniseikonia in objective and subjective measures of visual discomfort during prolonged reading tasks.

The objective measure of eye discomfort (*i.e.,* OO muscle activity) only reached statistical significance with a 10% degree of aniseikonia, whereas participants reported visual discomfort from the 3% degree of aniseikonia. These discrepancies in the level of aniseikonia at which visual discomfort occur may be explained by the fact that self-reported variables tend to overestimate the effect in comparison to objective measures. In this line, greater effects on subjective measures have been obtained for studies assessing visual discomfort with glare (*Shi et al., 2021*) or work stressors (*Frese & Zapf, 1988*). Another possible explanation is that participants were aware that the computer reading task was being performed with an afocal magnifier, and their subjective response could have been influenced by this bias.

The mechanisms linking visual fatigue to OO muscle activity remain an area of ongoing research. Investigating the connections between the OO muscle and fascial network pathways (Tenon's capsule: upper eyelid elevator, orbicularis oculi, sequentially superficial musculoaponeurotic system) can provide insights into aniseikonia and changes in OO muscle activity (*Zieliński et al., 2022*). The orbicularis oculi is a paired facial muscle that encircles each orbit and the adjacent periorbital region. Its intricate connections include merging with adjacent muscles, such as the levator labii superioris, levator nasolabialis, and zygomaticus minor. Additionally, peripheral fibers stretch into the temporal part of the epicranial aponeurosis. Recent studies have explored the relationship between muscle activity and eye characteristics, observing correlations between muscle bioelectrical activity and the length of the eyeball (*Zieliński et al., 2022*; *Zieliński et al., 2023*). Further research is needed to explore the intricate network and its impact on muscle function and eye health.Taken together, these findings evidence, using OO muscle activity and vision-related questionnaires, that relatively high levels of aniseikonia have a detrimental effect on visual fatigue. Nevertheless, there are several factors that may limit the generalizability of our findings, and they should be acknowledged. First, we measured the OO muscle activity in artificially induced aniseikonia rather than in patients with aniseikonia. It would be of interest to replicate this study in patients with different types of aniseikonia (*i.e.,* retinal, cortical and optical aniseikonia) to confirm the external validity of our results in real-world contexts. Second, the visual system is capable of considerable adaptation, so that it is possible that visual discomfort to the sudden imposition of aniseikonia in healthy young observers may not correspond to the response after a neuroadaptation period (*Krarup et al., 2020*).

## CONCLUSIONS

Our data showed that induced aniseikonia at high degrees (10%) increases the OO muscle activity during a 7-min reading task compared to a control condition and lower degrees of induced aniseikonia. The OO muscle activity was similar in dominant and non-dominant eyes, suggesting a binocular response to induced aniseikonia. There were changes in

reported levels of visual discomfort when the degree of induced aniseikonia was fairly low (*i.e.,* 3%). These outcomes may be of relevance to better understand the visual effects of aniseikonia.

### Funding
The authors received no funding for this work.

### Competing Interests
Beatriz Redondo and Jesús Vera are Academic Editors for PeerJ.

### Author Contributions
- Beatriz Redondo conceived and designed the experiments, performed the experiments, authored or reviewed drafts of the article, and approved the final draft.
- Jesus Vera conceived and designed the experiments, analyzed the data, prepared figures and/or tables, authored or reviewed drafts of the article, and approved the final draft.
- Rubén Molina performed the experiments, prepared figures and/or tables, authored or reviewed drafts of the article, and approved the final draft.
- Alejandro Molina-Molina conceived and designed the experiments, analyzed the data, authored or reviewed drafts of the article, and approved the final draft.
- Raimundo Jiménez conceived and designed the experiments, authored or reviewed drafts of the article, and approved the final draft.

### Human Ethics
The following information was supplied relating to ethical approvals (i.e., approving body and any reference numbers):

The experimental protocol followed the guidelines of the Declaration of Helsinki and was approved by the University of Granada Institutional Review Board (IRB approval: 546/CEIH/2018).

### Data Availability
The raw measurements are available in the Supplementary File.

### Supplemental Information
Supplemental information for this article can be found online at http://dx.doi.org/10.7717/peerj.17293#supplemental-information.

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
