# Peer review of "Orbicularis oculi muscle activity during computer reading under different degrees of artificially-induced aniseikonia"

_PeerJ, doi:10.7717/peerj.17293_

## Round 0.1 · original submission · Major Revisions

Both reviewers think this paper is interesting. However, both have a number of questions related to the methodology and the statistical analyses. Please carefully address each of these comments.

·

Basic reporting

no comment

Experimental design

no comment

Validity of the findings

no comment

Additional comments

This article presents an intriguing study on the impact of induced aniseikonia on visual fatigue. The paper adopts a novel approach and is, for the most part, clearly articulated. However, I have several concerns and questions:
1. The study examines 0%, 3%, 5%, and 10% degrees of aniseikonia. What principles guided the selection of these specific degrees?
2. What specific method was used to randomly assign the levels of aniseikonia and reading passages for each participant?
3. The manuscript mentions that "Participants wore soft contact lenses when necessary," but lacks detailed information on the refractive status and the specific methods of refractive correction for participants. This raises the following considerations: The methodology involves participants engaging in a dynamic reading task with stabilized heads. for participants with anisometropia who used framed glasses for correction, there's a potential risk of aniseikonia during eye movements. This deviation from the static condition used in the New Aniseikonia Test, where the line of sight moves off the optical center of the glasses, could introduce additional aniseikonia during the task. Therefore, reporting the specifics of participants' refractive status, including the existence or extent of any anisometropia and the form of correction used, is necessary.
4. Regarding the reading task, were the number of sentences, structure, and character count consistent across all passages? Even with a uniform font size, differences in these textual elements could impact the reading experience and potentially contribute to varying levels of fatigue. Additionally, it's important to consider whether these passages were similar in difficulty. Variations in the complexity of content could affect participants' reading speed due to their differing cognitive levels, which in turn might influence levels of fatigue. Regarding the duration of the reading task, how was it handled if a participant finished reading before the allotted 7-minute period? If they ended up just looking at the screen for the remaining time, this could potentially affect the measurements of OO muscle activity. Providing these details would be helpful for a comprehensive understanding of your study.
5. This manuscript asserts that the electrical activity of the OO muscle serves as an objective marker of visual fatigue, referencing relevant literature. However, the Discussion could be enhanced by elaborating on the mechanisms underlying the influence of visual fatigue on OO muscle activity.
6. The References listed on lines 318-320 and 321-323 indicate two separate articles, labeled as '2008a' and '2008b' respectively, yet both share the same DOI and bibliographic information. Please verify these references to ensure their accuracy.

Reviewer 2 ·

Basic reporting

no comment

Experimental design

L62-63 - Add newer citation / newer / reword the sentence. I consider it a mistake to cite data from 1946.
L70 –‘’ Bannon RE, 1944; Burian, 1946;’’ - Same situation. Update sentence and quotes.
L81 - How to describe the connections between the eyeball and the facial muscles of the cranium using the fascial network. Orbicularis oculi muscle is one of the muscles-fascia links of this network. This may explain the connection between aniseikonia and changes in muscle activity. Adding a description here can justify the study of Orbicularis oculi and in the information in the discussion can help to understand the results. Works may be helpful DOI: 10.3390/jcm12124166 ; DOI: 10.1038/s41598-023-47550-6

L99-100 – ‘’ Twenty-four collegiate students (mean age ± standard deviation: 24.00 ± 3.86 years; 16 female) were recruited to participate in this study.’’ - Prose add the calculation of the size of the sample.
L101-109 - The authors focus on the inclusion criteria from an ophthalmic perspective. This is completely correct, but in the work they also study the muscular system (Orbicularis oculi muscle). What inclusion and exclusion criteria were used in terms of the musculoskeletal system? In the studys reciprocal influence of the organ of vision on the musculoskeletal system has been demonstrated, so it is important to add additional information.
The work of the group of Dr. A. Monaco and Dr. G. Zielinski may be helpful here. I recommend familiarizing yourself with them.

L109-111 – ‘’ All participants were asked to abstain from alcohol and caffeine-based drinks 24 and 12 h before the experimental session, respectively, and to sleep at least 7 h the night prior to testing.’’ - What about medicines? (NPLZ or any medications that affect muscle activity?)
116 - Induced aniseikonia - I suggest that graphics with lenses used in the study would improve the clarity of the work.
139- ‘’ Muscle Activity Measurements and Signal Processing‘’ To this paragraph I have many questions and suggestions, all are in the following subsections:
1. At what times was the survey conducted? There is a diurnal variation in impedance during the electromyography study. The key is to determine the hours of the examination?
2. How many percent alcohol was used to clean the skin?
3. The authors used 5 × 5 cm Ag/AgCl electrodes, which are unusually large electrodes considering the muscle under study. How were the electrodes glued ? Please add additional graphics in the text with the location of the electrodes.
4. Ag/AgCl 5 × 5 cm, what was the conductive surface ?
5. Why was it decided to have 5 electrodes?
6. What was the impedance of the electrodes? Was it measured?
7. What was the sample rate? What is the input impedance of the used electromiography? What was the common mode rejection ratio?
8. What was the input range?, What was the baseline noise?
9. The protocols must be reproducible and the authors do not describe this part of the study enough.

L222 – DISCUSSION - In your discussion, add information about the anatomical connections between the systems under study.

Validity of the findings

L192 – ‘’ ANOVA’’ - As for the p-value for the ANOVA test. Bonferroni correction should be applied. DOI: 10.1111/opo.12131
L197-198 –‘’ The magnitude of the differences was reported by the partial eta squared (η²p) for Fs and Cohen's d effect size (ES) for t-tests, respectively (Cohen, 1988)’’ - This is too vestigial a description. Please write in detail what ranges were adopted for small, medium and large ES. How ES was calculated - what formulas were used.
Works may be helpful ‘’The need to report effect size estimates revisited. An overview of some recommended measures of effect size’’

Additional comments

Congratulates the authors on the idea of the work. It is indeed very interesting. However, it calls for improvement.

---

## Round 0.2 · accepted · Accept

Thank you for thoroughly addressing the reviewer comments. Both reviewers agree that the manuscript can be moved to the publication stage. However, please review all references to ensure uniformity and completeness.

·

Basic reporting

no comment

Experimental design

no comment

Validity of the findings

no comment

Additional comments

It is evident that great efforts have been made to address the concerns and recommendations that I raised in the initial review. The revisions have notably enhanced the coherence, rigor, and overall quality of the manuscript, making the arguments more compelling and the findings more robust and substantiated. Therefore, I recommend that the manuscript be accepted for publication in its current form.

Reviewer 2 ·

Basic reporting

no comment

Experimental design

no comment

Validity of the findings

no comment

Additional comments

The authors have responded correctly to my comments. I have no further substantive comments. I recommend the work for publication. Yours faithfully